# Detection Method for Walnut Shell-Kernel Separation Accuracy Based on Near-Infrared Spectroscopy

**DOI:** 10.3390/s22218301

**Published:** 2022-10-29

**Authors:** Minhui An, Chengmao Cao, Zhengmin Wu, Kun Luo

**Affiliations:** 1School of Engineering, Anhui Agricultural University, Hefei 230036, China; 2School of Tea and Food Science, Anhui Agricultural University, Hefei 230036, China

**Keywords:** walnut, NIRS, shell-kernel separation, SVM, ELM

## Abstract

In this study, Near-infrared (NIR) spectroscopy was adopted for the collection of 1200 spectra of three types of walnut materials after breaking the shells. A detection model of the walnut shell-kernel separation accuracy was established. The preprocessing method of de-trending (DT) was adopted. A classification model based on a support vector machine (SVM) and an extreme learning machine (ELM) was established with the principal component factor as the input variable. The effect of the penalty value (C) and kernel width (g) on the SVM model was discussed. The selection criteria of the number of hidden layer nodes (L) in the ELM model were studied, and a genetic algorithm (GA) was used to optimize the input layer weight (W)
and the hidden layer threshold value (B) of the ELM. The results revealed that the classification accuracy of SVM and ELM models for the shell, kernel, and chimera was 97.78% and 97.11%. The proposed method can serve as a reference for the detection of walnut shell-kernel separation accuracy.

## 1. Introduction

Walnuts are extremely nutritious nuts [1]. The preliminary processing of walnuts is mainly divided into the following steps: Green husk removal, walnut drying, walnut size classification, walnut shell-breaking, and walnut shell-kernel separation [2]. At present, some factories use manual sorting for the walnut shell-kernel separation, leading to increased labor costs and contamination of walnut kernels [3]. Effective, clean shell-kernel separation technology is critical for improving the quality and economic efficiency of the walnut industry. The traditional shell-kernel separation technology of nuts can be mainly divided into two categories-screened selection method [4] and the wind selection method [5]. The screened method uses the difference in particle size between the shell and the kernel to separate the material. The wind method is based on the different suspension speeds of the shell, kernel, and chimeras. Walnuts contain a Diaphragma juglandis Fructus (DJF) when compared with other nuts, which leads to the presence of shell-kernel chimera in the material after breaking the shell. Chimeras were when the kernel was sandwiched between the shell and the DJF (Figure 1). It increases the difficulty of walnut shell–kernel separation.

The NIR analysis method is simple, non-destructive, rapid, and accurate [6]. In recent years, NIR spectroscopy has been implemented in a wide range of applications for the analysis of agricultural products [7,8,9,10]. Few studies were conducted on the application of NIR spectrum instruments to walnuts, which were focused on the origin classification and composition detection of walnuts. In reference [11], partial least squares regression and support vector machine regression algorithms were used to predict the fat content of walnuts. In reference [12], fast prediction of antioxidant properties of walnut by NIR spectroscopy combined with multivariate analysis. There was no report on the problem of walnut shell kernel separation using NIR spectroscopy. Limited by its own experimental principle, NIR is not suitable for large-scale separation of walnut shell-kernel, but it can sample and detect the sorted materials to judge the separation accuracy of walnut shell–kernel. This provided a feasible scheme for on-line sampling and detection of walnut kernel separation accuracy.

In this study, NIR information from shells, kernels, and chimeras was analyzed. Spectral feature information was extracted by conducting principal component analysis (PCA). The effects of different preprocessing methods on the accuracy of the model were compared, and the parameter selection and optimization problems of the walnut shell-kernel separation effect detection model based on the support vector machine (SVM) and extreme learning machine (ELM) were investigated.

## 2. Materials and Methods

### 2.1. Materials and Instruments

The walnuts selected in the experiment were Anhui Ningguo Carya cathayensis Sarg with diameters of approximately 16–20 mm, clean surfaces, and no foreign matter. Carya cathayensis Sarg has a smaller volume and thicker DJF. More chimeras after the shell were broken compared with ordinary walnut. After the walnuts were broken using a professional shelling machine, 400 kernels, 400 shells, and 400 chimeras were selected as samples. All the samples were stored under ventilated, dry, and insect-free conditions. The spectrum acquisition instrument used was a FieldSpec4 ground object spectrometer (Analytica Spectra Devices., Inc, Boulder, CA 80301, USA). The spectrum data-processing software ViewSpecPro (Version 5.6, Analytica Spectra Devices., Inc, Boulder, CA 80301, USA), which was matched with the ASD spectrometer, was used to process the data.

### 2.2. Empirical Method

The NIR spectrum acquisition device built according to experimental requirements is shown in Figure 2. The device included a darkroom, a FieldSpec4 (with a spectrum acquisition range of 350–2500 nm and standard resolution of 1 nm), pistol handle, fiber optic cable, halogen lamp (50 W), and computer. The height and angle of the pistol handle were adjusted to avoid shadows and ensure that the sample was the only object to be measured in the field angle. The spectrum reference plate was collected at 30 min intervals to obtain the optimum signal-to-noise ratio of the spectrum data, and a black velvet cloth was used as the background in the darkroom.

The spectrum acquisition parameters were set as follows: the number of pre-average times of the white board was 10, the number of pre-average times of the sample was 10, and the number of average times of the dark current was 25. The spectrometer was warmed for 30 min before collection. The sample was placed at the center of the field, the optical fiber probe was set perpendicular to the sample, and the RS3 software was activated for spectrum collection. To reduce the influence of the storage time and environment temperature on the sample moisture, the spectrum collection of all the samples was completed within 24 h.

### 2.3. Spectra Data Preprocessing

The original spectra contain useful chemical information, significant background noise, and irrelevant information. It was necessary to remove the noise and uninformative variables from the spectra by using preprocessing methods. Multiplicative scatter correction (MSC), standard normal variate (SNV), de-trending (DT) were adopted to preprocess the original spectra of the three materials to eliminate interference. MSC was used to eliminate the scattering phenomenon caused by uneven particle distribution and particle size. SNV was used to eliminate the influence of solid particle size, surface scattering, and optical path transformation on diffuse reflection [13]. DT was used to eliminate the baseline drift of the spectra and was used after SNV. Four methods, MSC, SNV, DT, and SNV + DT, were chosen to preprocess the original spectra. The effects of the different methods on the accuracy of the models were compared.

The Kennard stone (KS) method was used to divide the data set. The ratio of the training set to the validation set was set at 5:3. For the three samples of shell, kernel, and chimera, 250 samples were extracted as the training set, and 150 samples were extracted as the validation set. There were a total of 750 samples in the training set and 450 samples in the validation set. The wavelength collection interval of the FieldSpec4 spectrometer was 350–2500 nm, the original training set was a matrix with dimensions of 750 × 2151, and the validation set was a matrix with dimensions of 450 × 2151. Principal component analysis (PCA) was adopted to reduce the dimensionality of the data [14]. PCA synthesizes high-dimensional variables into linear independent low-dimensional variables, which are called PCs. PCs explain the variance of the data matrix. These selected PCs can replace the original data when the cumulative explained variance reaches 95%.

### 2.4. Support Vector Machine Model

The underlying principle of the Support vector machine (SVM) is to separate various feature types in the feature space by searching for a specific hyperplane [15,16]. The objective of SVM is to minimize the cost function:(1)$$\frac{\lambda }{2}\left\|\omega^{2}\right\|+\left[\frac{1}{\eta }\sum_{i=1}^{n}\max\left(0,1-y\left(\omega X+b\right)\right)\right]$$

The first part of the cost function is referred to as the regularization term, and the second part is referred to as the loss term. Moreover, *λ* is removed from the regularization term and replaced with the penalty value *C*, which is multiplied by the loss term. The cost function is transformed into:(2)$$\frac{1}{2}\left\|\omega^{2}\right\|+C\left[\frac{1}{\eta }\sum_{i=1}^{n}\max\left(0,1-y\left(\omega X+b\right)\right)\right]$$

Adding kernel functions to traditional SVM models to solve non-linear problems. The RBF kernel function is widely used in SVM classification models. The expression of the RBF kernel function is:(3)$$K=\left(x_{i},x\right)=\exp\left(-{\left\|x_{i}-x\right\|}^{2}/2\sigma \right)$$

By replacing 1/2σ with g, this formula can be transformed into:(4)$$K=\left(x_{i},x\right)=\exp\left(-g{\left\|x_{i}-x\right\|}^{2}/2\sigma \right)$$

Two parameters need to be selected in this model, including the penalty value C and the kernel width g [17]. Exploring the effect of *C*, g parameters on SVM models. Find options for the choice of *C*, g parameters.

### 2.5. Extreme Learning Machine Classification Model

The Extreme learning machine (ELM)does not use an algorithm based on the gradient descent method [18]. The output ELM of a single hidden layer feedforward neural network is:(5)$$\begin{array}{l} f_{L}\left(x\right)=\sum_{i=1}^{L}\beta_{i}h_{i}\left(x\right)=H\left(x\right)\beta \\ h_{i}\left(x\right)=g\left(\omega_{i},b_{i},x\right) \end{array}$$

$g\left(\omega_{i},b_{i},x\right)\;$ is the activation function. $\omega_{i}$ is the weight of the input layer (W). $b_{i}$ is the threshold of the hidden layer (B). β is the output weight between the hidden layer and the output layer. Further, W and B are randomly generated and remain unchanged during the training process, and output layer node is determined. Therefore, only the activation function and the number of hidden layer nodes (L) need to be set in the ELM training process. There are many kinds of activation functions. In this experiment, sigmod function was used as the activation function of ELM model. Sigmod is easy to derive and suitable for feedforward neural network. The influence of different hidden layer nodes number on the accuracy of the model is discussed.

The random selection of W and B enables ELM to have the advantages of a rapid learning rate and high generalization performance. However, it also causes randomness in the model results. Exploring the role of L in improving this randomness. In addition to this, GA algorithm was used to select the best W and B to improve the stability of the model.

### 2.6. GA-ELM

The optimization of the values of W and B in the ELM model using the GA algorithm is divided into the following three steps: (1) Artificially set the initial parameters of the GA [19]. (2) Setting the fitness function. The predicted output error of the ELM on the sample is used as the fitness function to calculate the fitness value of individuals in the initial population. (3) The better individuals are optimized using selection, crossover, and mutation to obtain new populations. The operation ends when the maximum number of iterations is reached. Select the optimal population individuals and identify them as the optimal W and B. Complete the optimization of the ELM model.

### 2.7. Evaluation of Model Performance

*Accuracy* is used to evaluate model performance. The formula is:(6)$$Accuracy=\frac{n}{m}\times 100\%$$
*n* is the number of predicted and actual values that are equal. *m* is the number of actual values.

## 3. Results and Discussion

### 3.1. Spectra Data Preprocessing

The NIR absorption band is mainly formed by the vibration of different groups of molecules in organic matter. The main components of walnut kernels are fat, protein, water, and sugar. The main components of walnut shells are lignin, cellulose, and hemicellulose. The main components of DJF are amino acids, proteins, and flavonoids. Figure 3 presents the original spectra of the samples. The blue, red, and black lines indicate the spectra of the kernel, shell, and chimera, respectively. The absorption peak intensities of the three sample types were significantly different. In the wavelength range of 750–2500 nm, the order of the overall relative reflectance values was as follows: kernel > shell > chimera. There are overlaps in the spectra of the three materials. The overlap is not conducive to model classification. Therefore, further preprocessing should be conducted.

MSC, SNV, DT, and SNV + DT were adopted to preprocess the original spectra of the three materials to eliminate interference. The classification model based on ELM was established for the preprocessed full-band spectra. Figure 4 presents the model classification results of the different preprocessing methods. The DT preprocessing results were superior. The accuracy of the training sets reached 75.07%. The accuracy of the validation sets reached 75.09%. The results of MSC preprocessing were slightly poorer, and the results of the SNV and SNV + DT preprocessing were the poorest. DT was used as a preprocessing method. Figure 5 shows the spectra after different preprocessing.

### 3.2. Principal Component Analysis Dimensionality Reduction

Dimension reduction of spectra data was carried out by using PCA. The results are shown in Table 1.

The variance contribution rates of the first six PCs were 47.07%, 36.92%, 7.04%, 4.09%, 1.75%, and 1.04%, respectively. Figure 6 displays the projection of the validation set in the PCA space built with the training set. Different types of samples tended to exhibit trends of separation, and samples of the same type exhibited significant aggregation. The cumulative contribution rate of the first PC (PC1) and second PC (PC2) was 83.99%, which was not enough to fully explain all the information of the original data. The cumulative contribution rates of the first six PC was 97.91%. The first six PCs were extracted for the classification model training [20].

### 3.3. Model Development: SVM

There were two parameters that needed to be selected in this model, including the penalty value C and the kernel width g [21]. Moreover, C results in a trade-off between the error classification samples and the simplicity of the classification surface. High values of C impose high penalties on errors and make the model more rigorous, thus increasing the bias and reducing variance. Low values of C result in a high tolerance to errors, which expands the boundary. However, the variance is increased. g could be considered as the reciprocal of the radius of influence of the samples selected by the SVM model, which affects the model complexity. This experiment explored the ideas for finding the optimal C and g parameters. The optimal C and g were also determined. The results are shown in Table 2.

In experiments 1–5, g was set as 0.01, and C was set as 1, 2, 3, 4, and 5, respectively. The accuracies of the training set were 98.27%, 99.47%, 99.73%, 100%, and 100%. As the value of C increased, the model exhibited a greater penalty for errors, the size of the boundary surface decreased, and the tolerance for errors decreased. For the training set to be accurately classified, a large C value was required. However, an excessively large C value posed a high risk of overfitting. The accuracies of the validation set decreased as the value of C increased. In experiments 5–10, with a C value of 5, the influence of the g value on the model was investigated. In particular, g was decreased from 0.01 to 0.00005, The accuracies of the validation set changed from 90.22% to 91.33%, 95.11%, 97.11%, 97.78%, and 97.56%. The accuracies of the training set decreased from 100% to 99.47%, 97.2%, 96.8%, 95.73%, and 95.07%. In this model, the optimal values of C and g were C = 5 and g = 0.0001. The accuracies of the training set and validation set were 95.73% and 97.78%.

### 3.4. Model Development: ELM

An ELM classification model with sigmod function as activation function was established. Experiments were carried out to explore the selection criteria of the number of hidden layer nodes (L). Given that the weight of the input layer and the threshold of the hidden layer of the ELM were randomly selected, the experimental results exhibited features of randomness. The average value extraction method was used to reduce the contingency of the model in multiple experiments. If the L is excessively large or small, the learning ability and complexity of the network structure are affected. At present, there are several empirical methods for selecting L, which lack theoretical support. The L obtained is, therefore, for reference only. Table 3 presents the training and validation results of the ELM model with different values of L.

In Experiment 1, according to Fangfagorman’s theorem, the relationship between L and the number of modes N is L = log2 N. The value of N was 750. Therefore, L = 10. The average accuracy of the training set was 85.69%. The average accuracy of the validation set was 82.37%. In Experiment 2, Kolmogorov highlighted that the L can be expressed as L = 2n + 1 (where n is the number of input layer nodes). Six principal components were extracted from the data using PCA dimension reduction. Therefore, L = 13 for *n* = 6. The average accuracy of the training set was 87.78%, and the average accuracy of the validation set was 87.11%. L was excessively small, and the network learning ability was insufficient. In Experiments 3–8, the accuracy of the training set increased as L increased. For L = 70, the average accuracy of the training set was 94.31%. However, the accuracy of the validation set increased first and then decreased as L increased. The highest accuracy of the validation set was achieved when the value of L was 40. The highest accuracy rate was 93.04%. The L of the ELM model was set to 40 to avoid overfitting. The average accuracy of the training set and validation set were 92.89% and 93.04%, respectively.

GA was adopted to optimize W and B of the ELM to create the optimal network [22]. The model comparison results of the GA-ELM and ELM are shown in Table 4. It can be seen from the results that the GA determined the optimal W and B under the current L and improved the identification accuracy of the ELM. The accuracy of the training set and validation set in the ELM model optimized by GA was 95.87% and 97.11% when L was 40.

## 4. Conclusions

In this study, the separation accuracy of walnut shell and kernel was detected by NIR spectroscopy. A shell kernel classification model was established based on an SVM and ELM. A comparison of the influences of different preprocessing methods on the accuracy of model recognition was presented. DT can effectively remove spectra changes caused by the baseline drift, and the obtained walnut shell kernel classification model was the best. Dimensionality reduction was achieved by conducting PCA. The first six PCs were extracted for the classification model training.

The method of choosing the C and g parameters in the SVM was explored. The accuracy of the training set can be improved by increasing C, but too large a value of C will lead to overfitting. The accuracy of the validating set can be improved by decreasing g. However, the decrease of g is limited. Too small g will reduce the accuracy of the training set and verification set. The selection criteria of L were explored in the ELM. Increasing L can improve model accuracy by increasing the learning capability of the network. Too large L will lead to overfitting and reduce the accuracy of the validation set. To improve the uncertainty caused by the random selection of W and B. The GA algorithm was used to select the best W and B.

Overall, the shell-kernel classification model based on SVM and ELM developed in this study has 97.78% and 97.11% accuracy for the validation set. It can effectively evaluate and detect the accuracy of the existing shell-kernel separation method.

## Figures and Tables

**Figure 1 sensors-22-08301-f001:**
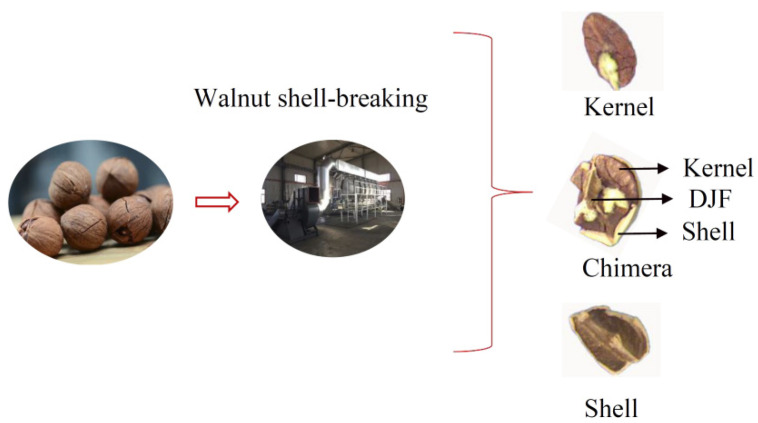
The phenomenon of misclassification after walnut shell–kernel separation.

**Figure 2 sensors-22-08301-f002:**
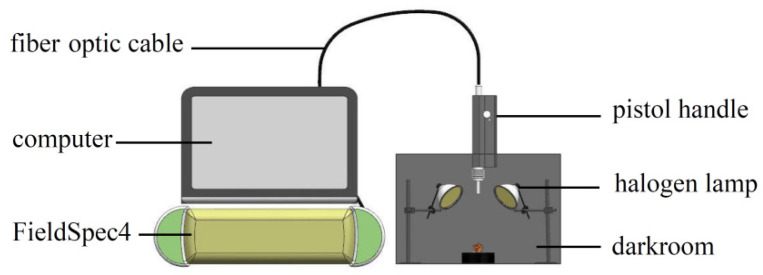
Spectrum acquisition test bench.

**Figure 3 sensors-22-08301-f003:**
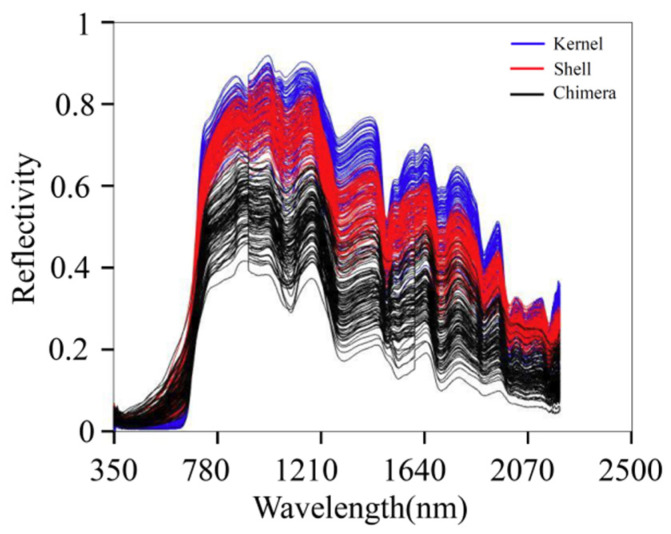
Original spectra.

**Figure 4 sensors-22-08301-f004:**
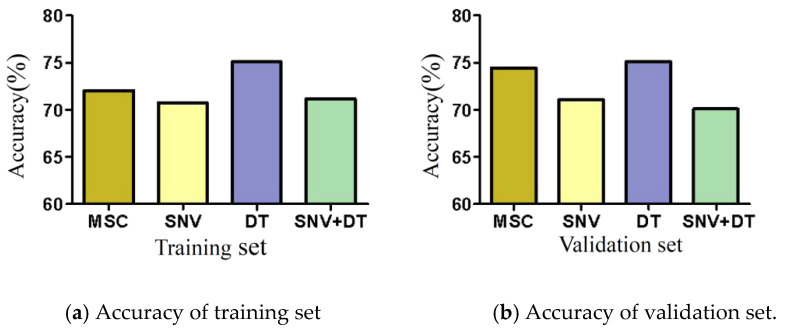
Model classification accuracy of different preprocessing methods.

**Figure 5 sensors-22-08301-f005:**
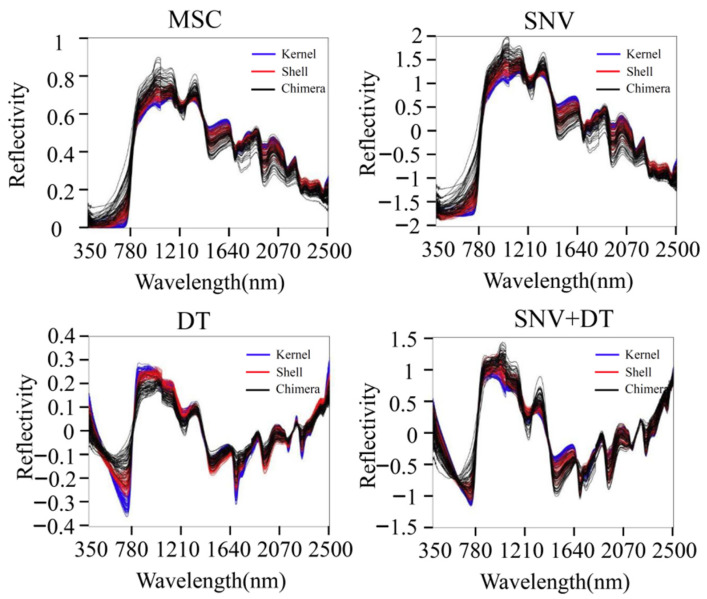
Spectra after different preprocessing methods.

**Figure 6 sensors-22-08301-f006:**
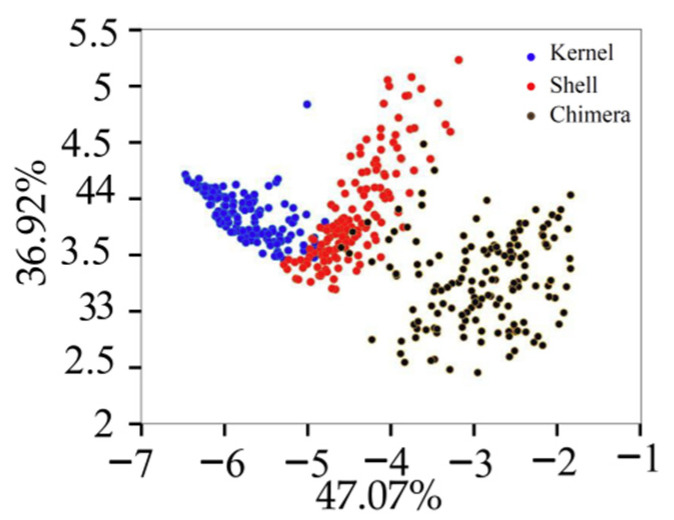
Projection of the validation set in the PCA.

**Table 1 sensors-22-08301-t001:** Results of Principal Component Analysis (PCA).

PC	Eigenvalue	Variance Contribution (%)	Cumulative Contribution (%)
1	1012.47	47.07	47.07
2	749.1142	36.92	83.99
3	151.42	7.04	91.03
4	87.95	4.09	95.12
5	37.55	1.75	96.87
6	22.44	1.04	97.91

**Table 2 sensors-22-08301-t002:** Results of SVM model with different C and g parameters.

Experiment	C and g	Training Set		Validation Set	
		Accuracy (%)	Misjudgment	Accuracy (%)	Misjudgment
1	C = 1 g = 0.01	98.27	13	94.22	26
2	C = 2 g = 0.01	99.47	4	93.33	30
3	C = 3 g = 0.01	99.73	2	92.22	35
4	C = 4 g = 0.01	100	0	91.33	39
5	C = 5 g = 0.01	100	0	90.22	44
6	C = 5 g = 0.005	99.47	4	91.33	39
7	C = 5 g = 0.001	97.2	21	95.11	22
8	C = 5 g = 0.0005	96.8	24	97.11	13
9	C = 5 g = 0.0001	95.73	32	97.78	10
10	C = 5 g = 0.00005	95.07	37	97.56	11

**Table 3 sensors-22-08301-t003:** Results of ELM model with different values of L.

Experiment	L	Dataset	First (%)	Second (%)	Third (%)	Average (%)
1	10	Training	87.07	82.80	87.20	85.69
		Validation	82.44	79.78	84.89	82.37
2	13	Training	87.87	87.33	88.13	87.78
		Validation	86.89	86.89	87.56	87.11
3	20	Training	86.60	87.07	90.40	89.02
		Validation	88.67	85.11	88.44	87.41
4	30	Training	92.27	91.87	92.27	92.13
		Validation	90.89	89.33	88.67	89.63
5	40	Training	92.80	93.60	92.27	92.89
		Validation	92.44	94.89	91.78	93.04
6	50	Training	95.07	93.07	93.47	93.87
		Validation	94.00	92.00	89.78	91.93
7	60	Training	93.20	94.40	94.27	93.96
		Validation	91.33	93.78	89.56	91.56
8	70	Training	94.80	94.13	94.00	94.31
		Validation	88.89	91.11	93.33	91.11

**Table 4 sensors-22-08301-t004:** Results of comparison between the GA-ELM and ELM models.

L	Accuracy of Training Set (%)		Accuracy of Validation Set (%)	
	ELM	GA-ELM	ELM	GA-ELM
30	92.13	94.93	89.63	96.89
40	92.89	95.87	93.04	97.11

## Data Availability

Not applicable.

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
