# Peer review of "Detection Method for Walnut Shell-Kernel Separation Accuracy Based on Near-Infrared Spectroscopy"

_sensors, 2022, doi:10.3390/s22218301_

Round 1

Reviewer 1 Report

Detection method for walnut shell-kernel separation accuracy 2 based on Near-infrared spectroscopy

This is an interesting paper. I have following observations:

Results and Discussion:

In general, the results are only described. More detailed discussion and bibliographic support is needed. What was the puurpose of using this specific pretreatment (MSC), there are many other options as well, please justify. 

Some minor comments:

Line 14 – What is C, g here ? Please avoid use of abbreviations in the abstarct

Line 27- Replace “intelligent” with “effective/ smart”

Line 43- Don’t repeat “In reference” again and again

Line 61- Rephrase the line

Line 70- Comma after volume

Line 71- In the following, (cut line)

81- Replace “is” with “an”

Line 108- Dividing. (Rephrase line)

Line 123- Remove “to” from referred to

Line 124- Remove “to” from referred to

Line 128- Un-conspicuously

Line 163- Remove dot after figure

Line 165- The capital

Line 166- Remove dot and use “and”

Line 172- Remove dot from figure.5

Line 184 - Was done by using

Line 234- Replace “impacted” with affected

Line 237- Different value of L

Line 265- Remove dot from table.4

Line 270- The “t” small letter

Line 275 - The “t” small letter

Spacing issues/ Tables alignment/ Missing Issue number in some references (6, 7, 8, 9, 10, 12, 13, 14, 17, 18)

Reviewer 2 Report

General comment: The authors carried classification of kernel, shell and chimera of walnut using PCA, SVM and ELM. The methods section lacks critical information and discussion could have been elaborated.

Following comments may be addressed

L24: ……following steps: green husk……..

L35: Check spacing. kernel#(Figure 1)

L122, L126, L137 Re-write the equations

L142: ELM

For 2.4 and 2.5 sections, the implementation was not provided. So explain how both SVM and ELM were implemented. Also cite appropriate references for the algorithms. Huang et al. 2011 is missing.

In-house developed code or used public computational routines?

L163: Check period. It is Figure 4, not Figure. 4

L166, 172: Check period

Figure 4: How is the percentage accuracy measured using ELM. Provide the formula or equation for the same in methods.

Entire manuscript, correct Figure.X to Figure X

L175: Separation degree? What does it mean? How to calculate this metric?

L187: Check case

L204-214: It is essentially grid search for C and gamma, which is common during SVM regression/classification. It doesn’t need a separate result table.

What is value of regulation parameter (‘C’) in ELM?

Did you optimize it as well or simply fixed it to some value?

Figure 7 is re-representation of Table 3?

L263-267: The implementation of GA was described in methods

L278-279: ‘g’ or ‘G’

Reviewer 3 Report

The authors have developed a classification method to detect and separate walnut shell-kernels based on Near-infrared spectroscopy. The problem is interesting, although a comment on how to implement it at an industrial level should be included.

The paper needs an extensive revision of the English language. The experimental procedure is, in general, well described, and the results seem good.

However, my main concern is about preprocessing of the spectra and the need to use so sophisticated classification methods such as SVM and ELM.

I am not sure the MSC preprocessing is applied correctly, as it provides very different results from other preprocessing techniques. In fact, MSC corrected spectra are just so different (for the 3 types of samples) that you do not need to apply any classification method. Just plotting the spectrum of a new sample and comparing would be enough. The same with the PCA. Classes are grouped very clearly, so the need for SVM and ELM is in doubt.

My specific remarks are included in the pdf file attached to this revision.

Reviewer 4 Report

The paper presents the application of NIR technology for improving the separation of shell-kernel of nuts. Authors use a large dataset and different pretreatment methods to select the best model. The results seem to be useful for the proposal. However, some information is missing so that the document is difficult to follow. Concretely the section Material and Methods should be extended explaining better the experiments and the methodology used.

In line 62, the authors indicate that “different preprocessing methods on the accuracy of the model were compared and the parameter selection and optimization problems of walnut shell-kernel separation effect detection model based on the support vector machine (SVM) and extreme learning machine (ELM) were investigated”

But, really, it is difficult to follow which are the different preprocessing methods and how the parameters are selected and which are these parameters (may be they are C and g), but this should be explained in material and methods.

Comments like those include in lines 129-132, are not useful, it should be explained directly what authors did not and what can be done. The same comment is valid for lines 134-135, please avoid the use of what does not use and change by what was used.

Other example is in Line 142-143: “There are many kinds of activation functions”. This sentence is not useful to indicate which activation function was used. It you use a sigmoid function, explain the reasons in material and methods. Note that  lines 228-229 mention this again.

Table 2 and 3 refer to different experiments, but the experiments are not explained before in material and methods. The same about parameters as C (layers node) or g. Note that in the text there are “g” and “G”.

In my opinion, this is the main weakness of this paper. Material and methods section needs to be improved to understand what authors did.

Other comments:

In Introduction, lines 43-54. There are many references regarding the use of NIR technology for agro food products, so the paragraphs can be easily improved. Then the paper provides more precise information about the use of NIR and its applications. Please do not refer to a specific reference unless it refers to specific data related to your paper. It seems more interesting for readers to refer to the application of NIR technology such as composition, antioxidant properties or maturity for example.  

The quality of Figure 1 should be improved. English is not my mother tongue, but I am not sure if the nuts in the photos are walnut or hazelnut.

Lines 70-72 are not understandable.

Please explain the meaning when you are talking about chimaera.

Round 2

Reviewer 2 Report

Overall it is fine modification.

Kindly check the typographical errors such as repetition of conclusion section. 

Reviewer 4 Report

There are minor mistakes in the text that can be corrected during edition.

Line 35, space before NIR

Line 129, space before exploring

Line 141, space before In this experiment

Lines 151-162 several spaces after points.

Line 253, delete "In expe"

Line 269

Several more spaces

Line 286. Delete 4. Con
